# Insights into the viral landscape of the western honey bee and native bees in Bangladesh

Islam Hamim,[1] Lena Wilfert[2]

**ABSTRACT** Bees are important pollinators that are increasingly threatened by viruses. In this study, we investigated the viruses in honey bees in Bangladesh, focusing on western (*Apis mellifera*) and native bee species (*A. cerana*, *A. dorsata*, *A. florea*, and *Trigona* sp.). Using high-throughput poly(A)-selected RNA sequencing, we observed that viruses of the order *Picornavirales* are frequently detected in both western and native bees. However, this pattern may reflect both true biological abundance and methodological bias, as this approach inherently enriches for polyadenylated RNA viruses. Deformed wing virus (DWV), black queen cell virus (BQCV), and sacbrood virus (SBV) were commonly found in western bees, while native bees exhibited a high diversity of viral communities rather than dominance of specific viruses. The common bee viruses also showed high read abundances in western bees. Notably, the study identified unreported viruses in bees belonging to the Iflaviridae and Dicistroviridae families, expanding the known diversity of honey bee pathogens. In addition, plant-associated viruses were identified, suggesting a potential role for honey bees as vectors of plant viruses and highlighting the interactions between bees, plants, and their pathogens. The results of the diversity analysis demonstrated significant differences in the composition of virus populations between western and native bees in our studied samples. These results reveal the occurrence of bee viruses in Bangladesh and highlight the potential interspecific transmission of viruses, which may pose a significant threat to local bee populations. Our study emphasizes the importance of monitoring known viruses and novel viruses, as well as plant pathogens, and implementing sustainable management practices to reduce the spread of pathogens and protect both native and western bees.

**IMPORTANCE** Pollinators face increasing threats from viral pathogens, yet data on their viromes remain limited in many parts of the world, including South Asia. This study provides insights into the viral communities of both native and non-native bee species in Bangladesh using RNA sequencing. While *Apis mellifera* showed higher viral loads of known honey bee viruses, native bee species exhibited a broader diversity of viral sequences, including several uncharacterized viruses. Although based on a limited sample set, these findings contribute to a growing understanding of viral diversity in pollinators and underscore the value of continued surveillance to better understand virus-host associations and potential cross-species transmission in regions undergoing rapid apicultural expansion.

**KEYWORDS** virology, environmental microbiology, plant pathology, bee pathology

Bees are essential pollinators of agricultural crops and flowering plants, contributing to the conservation of biodiversity and agro-ecosystems. Furthermore, both the western honey bee (*Apis mellifera*), native to Africa and Europe, and the eastern honey bee (*A. cerana*), endemic to Asia, produce honey and other substances important for human consumption. Bee populations have declined significantly worldwide, often

**Peer Reviewers** Ivan Toplak, Veterinary Faculty, University of Ljubljana, Ljubljana, Slovenia; Dominika Kadleckov, Charles University, Prague, Czech Republic

Address correspondence to Islam Hamim, hamimppath@bau.edu.bd.

The authors declare no conflict of interest.

resulting in reduced agricultural production due to inadequate pollinator visits (1, 2). This has implications for our food supply, economic stability, biodiversity, and ecosystem functioning. The main causes of decline in pollinator and bee populations are habitat loss, agrochemical use, climate change, pests, and pathogens (3, 4). In managed bees, particularly in *A. mellifera*, emerging and widespread diseases are important threats to the health of bee populations and the sustainability of beekeeping (5). Despite the importance of bee pathogens, research is lacking in large parts of the world, including in South and Southeast Asia.

In western populations of the honey bee *A. mellifera*, viruses have been intensively studied because of their economic importance. Viruses infecting honey bees are linked to a variety of symptoms, ranging from asymptomatic infections to severe mortality and colony collapse. Although most bee viruses cause latent infections, some viruses, such as sacbrood virus (SBV), black queen cell virus (BQCV), and deformed wing virus (DWV), cause apparent symptoms and death of individual bees (6). Members of the DWV and acute bee paralysis virus (ABPV) clades are also associated with the collapse of entire colonies of *A. mellifera*, particularly in the presence of the ectoparasitic mite *Varroa*. *A. cerana*, which is of comparable apicultural importance to *A. mellifera* in Asia, typically shows lower levels of virus infection, likely due to reduced *Varroa destructor* infestation (7–10). This species has co-evolved with *V. destructor* and employs a range of behavioral and physiological strategies to limit mite reproduction (11). In managed *A. cerana*, SBV is the main pathogen of concern, with outbreaks across Asia causing high mortality (9). For example, SBV infections in *A. cerana* led to the collapse of approximately 80% of colonies and a significant reduction in honey production in Bangladesh in 1990 (12).

Asia is the biodiversity hotspot for honey bees, with several wild honey bee species including the cave-dwelling *A. cerana*. This species occurs in wild populations but has also been kept by beekeepers for centuries. Additionally, managed apiaries of non-native *A. mellifera* are also common across Asia. In addition, many wild solitary and social bee species, from stingless bees to bumble bees, occur in Asia. Even though the eastern honey bee is generally less affected by infectious diseases than its western counterpart, Asia has been the origin of two recently emerged parasites: both the virus-vectoring *V. destructor* and the microsporidian *Nosema ceranae* jumped from *A. cerana* to non-native western honey bees in the last century and have since been anthropogenically spread around most of the globe (13, 14). With the recent advances in meta-transcriptomics, it has also become clear that honey bees harbor a high number of potential viral pathogens, with over 80 viral genomes now identified in honey bees (15), with more being constantly added (15, 16). The high risk of disease emergence and the lack of studies in South and Southeast Asia highlight the need for *de novo* meta-transcriptome-based studies in these regions.

Bangladesh is a country that exemplifies this need for meta-transcriptomic studies. It is home to several honey bee species, including wild bees, such as *A. dorsata* (giant honey bee) and *A. florea* (dwarf honey bee). The native species *A. cerana* (eastern honey bee) has traditionally been used for honey production under partially managed conditions, due to its adaptability to local conditions (12). However, in the early 1990s, *A. mellifera* was introduced to Bangladesh, following the collapse of *A. cerana* populations due to an SBV outbreak, and it has since become the dominant species for commercial beekeeping due to its higher honey yields (12). Honey bees in Bangladesh forage on a variety of crops, such as mustard (*Brassica campestris*), litchi (*Litchi chinensis*), black cumin (*Nigella sativa*), coriander (*Coriandrum sativum*), niger (*Guizotia abyssinica*), and sesame (*Sesamum indicum*), as well as wild plants in Sundarbans and the hilly regions of Chittagong (12). Honey is produced mainly in winter, using *A. mellifera* and *A. cerana*, while only a limited amount of honey is harvested from wild bees (17). The country has yet to commercialize pollination services as a distinct industry. However, the beekeeping sector, which is primarily focused on honey production for domestic consumption, plays an important role in increasing agricultural productivity by increasing crop yields through the provision of pollination services (12) and can provide significant benefits

for poverty alleviation and rural development through low-investment, high-return opportunities (17). However, honey bee health in much of Asia, including in Bangladesh, has received limited research attention, particularly with respect to virome structure and virus transmission.

The high diversity of honey bees and stingless bees in Southeast Asia and Bangladesh, and the overlap with the non-native western honey bee *A. mellifera*, may facilitate the interspecific transmission of pathogens and parasites (8). The spread of the virus-vectoring *Varroa* mite in *A. mellifera* has led to the spillover of DWV into wild bee species (18, 19). For example, research in China has shown that *A. mellifera* is a current source of DWV for *A. cerana* (10), and the spillover of pathogens from managed *A. mellifera* to wild species can cause harm, such as in stingless bees (20). Here, we used meta-transcriptomics to explore the occurrence and diversity of both known and novel viruses in bee species from Bangladesh. This approach allows us to assess not only the viruses that directly affect bees, but also those they may potentially transmit from plants or other environments. The results of this study underscore the importance of applying caution in beekeeping practices with *A. mellifera*, as this species appears to harbor a higher abundance of specific groups of potentially pathogenic viruses compared to native bee species such as *A. cerana*.

## RESULTS

### Viral load distribution in bee species

In this study, we used high-throughput, poly(A)-selected RNA sequencing (RNA-Seq) on the DNBSEQ platform (BGI, China) to investigate the viral landscape across different honey bee species in Bangladesh. Additionally, we were able to include one library each of stingless bees (*Trigona* sp.) and the small hive beetle *Aethina tumida*, an inquiline of the western honey bee *A. mellifera*. A total of 15 libraries were examined that include *A. mellifera* (eight libraries), *A. cerana* (three libraries), *A. dorsata* (one library), *A. florea* (one library), *Trigona* sp. (one library), and *A. tumida* (one library) (Fig. 1; Table 1). These included two libraries available from GenBank: one of *A. mellifera* and the other of *A. cerana* in Tangail, Bangladesh (NCBI BioProject PRJNA1090432). On average, 8.1% of the total reads were of viral origin (Table 1).

Western honey bees showed a range of viral read percentages from 0.067% to 69.777% (mean = 11.686%, median = 2.774, standard deviation = ±23.785) (Table 1), indicating high variability in viral loads across libraries. Native bees showed low viral load percentages ranging from 0.0007% to 0.1859% (mean = 0.0396%, median = 0.0081 and standard deviation = ±0.073), indicating an overall low viral load with less variation across the samples. The percentage of viral reads differed significantly among insect groups, defined here as combining bee and non-bee insects in this study ($\chi^2$ = 10.334, d.f. = 2, $P$ = 0.006), with higher viral read percentages in western honey bees. Dunn's test also confirmed a statistically significant difference ($P$ = 0.003) in the percentage of viral reads between native and western bees.

### Bee virome composition

The transcriptome analysis of bee viromes led to the identification and assembly of 50 viral operational taxonomic units (OTUs), including 10 putative plant viruses, through sequence homology. Using sequence homology and phylogenetic analysis, we classified the viruses into insect and plant virus families (Fig. 2; Table S1). Of these, viral families of the order *Picornavirales* were the most abundant across all bee viromes. This pattern may reflect both true biological abundance and methodological bias, as we used poly(A)-selected RNA sequencing approach that inherently enriches for polyadenylated RNA viruses and can exclude other viral groups. However, Iflaviridae was found as the most abundant family, with 95,098.55 reads per million (RPM), followed by Dicistroviridae (7,718.257 RPM) (Fig. 2; Table S1). Dicistroviridae and Iflaviridae were detected in all western honey bee libraries and in the *A. florea* library. Additionally, the Dicistroviridae

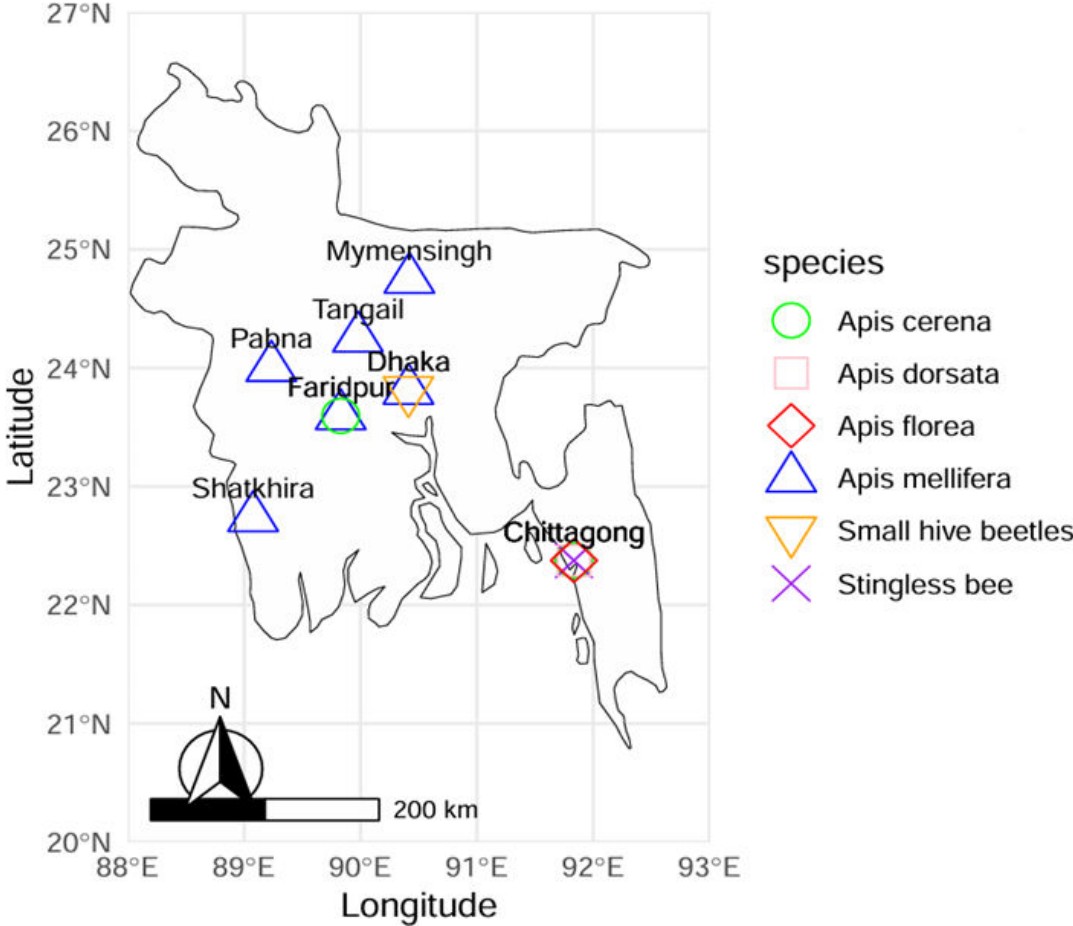

**FIG 1** Map showing sampling sites of five bee species (*A. mellifera, A. cerana, A. dorsata, A. florea, Trigona* sp./*stingless bees*) and an inquiline of honey bee, *Aethina tumida,* across Bangladesh. In total, 13 RNAseq data sets were produced from different samples of seven districts (Mymensingh, Pabna, Faridpur, Dhaka, Satkhira, Tangail, and Chittagong). Locations with overlapping species are distinguished by different colors and shapes representing each species.

were detected in two out of three *A. cerana* libraries and in A. *dorsata*, while Iflaviridae were detected in one of three *A. cerana* libraries. Neither was detected in the Meliponini library. Sinhaliviridae, with 1,062.95 RPM, was detected in five of eight *A. mellifera* libraries, indicating that its presence is restricted to western honey bees (Fig. 2; Table S1). Besides these, unclassified insect viruses were found in six of the libraries, including four native bee libraries, with a relatively low abundance (43.95 RPM). The family Secoviridae, typically associated with plant viruses, was found in three of the studied libraries, including the Meliponini (Fig. 2). Virus families, such as Alphaflexiviridae, Marnaviridae, and Fusaviridae were the rarest, each appearing in only one library.

## Identification of the phylogenetic positions of insect viruses

We identified the phylogenetic positions of the viruses by comparing the positions of virus sequences in the constructed evolutionary trees relative to known virus species and strains. The findings indicated well-defined monophyletic clades within the Iflaviridae family, including DWV, SBV, and *Varroa destructor* virus 2 (VDV2) (Fig. 3). These clades have strong bootstrap support and represent established lineages. On the Iflaviridae tree (Fig. 3A), DWV-A, DWV-B, DWV-C, and DWV-D form a monophyletic group, with both DWV-A and DWV-B present in Bangladesh, but no evidence of strains C or D (Fig. 3A). The SBV BD (Bangladesh) isolate clustered with an SBV isolate from Australia, while the VDV 2 BD isolate clustered with an isolate from China (Fig. 3A). In addition to these well-studied clades, the study identified unknown isolates of Iflaviridae collected in Bangladesh (Fig.

TABLE 1 Viral load distribution in bee species in Bangladesh: total and percentage of viral reads

| Library name | Host name | Group | Location | Total reads | %Viral reads |
|---|---|---|---|---|---|
| A_mellifera_Mymensingh_S | *A. mellifera* | Western | Mymensingh | 120,133,210 | 1.848 |
| A_mellifera_Mymensingh_M | *A. mellifera* | Western | Mymensingh | 110,423,366 | 12.221 |
| A_mellifera_Pabna | *A. mellifera* | Western | Pabna | 120,163,106 | 0.067 |
| A_mellifera_Faridpur | *A. mellifera* | Western | Faridpur | 33,615,952 | 1.224 |
| A_mellifera_Dhaka | *A. mellifera* | Western | Dhaka | 60,243,004 | 3.699 |
| A_mellifera_Satkhira | *A. mellifera* | Western | Satkhira | 120,176,902 | 69.777 |
| A_mellifera_Tangail | *A. mellifera* | Western | Tangail | 56,415,778 | 0.744 |
| A_mellifera_Tangail_1 | *A. mellifera* | Western | Tangail | 33,761,388 | 3.905 |
| Apis_cerana_Tangail_1 | *A. cerana* | Native | Tangail | 33,465,047 | 0.00072 |
| Apis_cerana_Faridpur | *A. cerana* | Native | Faridpur | 65,522,594 | 0.012 |
| Apis_cerana_Chittagong | *A. cerana* | Native | Chittagong | 120,170,831 | 0.034 |
| A_dorsata_Chittagong | *A. dorsata* | Native | Chittagong | 54,204,944 | 0.1859 |
| Apis_florea_Chittagong | *A. florea* | Native | Chittagong | 120,166,790 | 0.004 |
| Meliponini_Chittagong | *Trigona* sp. (Meliponini) | Native | Chittagong | 120,162,187 | 0.001 |
| Aethina_tumida_Dhaka | *Aethina tumida* | Other | Dhaka | 120,371,234 | 0.0001 |

3A). These unknown isolates could be new virus species or strains, showing new lineages within the family. The unknown isolates Bee Iflavirus BD 9 and Bee Iflavirus BD 7 were found as a potential outgroup to DWV (Fig. 3A; Fig. S1). Bee Iflavirus BD 7 and Bee Iflavirus BD 9 share 44% sequence similarity at the nucleotide level, while their similarity to DWV ranges from 58% to 68% and 46% to 58%, respectively (Supplemental material S1).

The phylogenetic analysis also revealed strong genetic relationships among various Dicistroviridae viruses (Fig. 3B). Here, BQCV BD clusters most closely with BQCV isolates from Pakistan. For the Sinhaliviridae, we found LSV SA2, LSV 3, and LSV 4 in Bangladesh (Fig. 3C), while the strains LSV 1 and LSV 2 were not found in our samples. In the LSV 4 group, LSV 4 BD is closely related to the previously reported isolates from Bangladesh and Pakistan, indicating a potential regional connection to South Asia. Within the LSV SA2 lineage, LSV SA2 BD is genetically distinct, forming a separate branch. Similarly, LSV 3 BD forms a branch of the LSV3 lineage, which is associated with Chinese and global isolates.

## Distribution of insect viruses

Among the insect viruses identified, the most prevalent virus species in bee samples studied were DWV-A, BQCV, bee-associated cripavirus 1, SBV, and bee dicistrovirus 1 (*Dicistroviridae* sp.). While DWV-A was absent in all native bee libraries, it was present in all eight western honey bee libraries (mean = 2,136.298 RPM, median = 563.967 RPM, standard deviation = ±3,158.03 RPM), four of which also contained DWV-B (mean = 31.876 RPM, median = 18.36 RPM, standard deviation = ±32.56 RPM) (Fig. 4). While LSV variants were only found in western honey bees, BQCV, SBV, bee-associated cripavirus 1, bee dicistrovirus BD 1 and 3, and *Planococcus ficus*-associated dicistrovirus 1 occurred in several western and native bee libraries. In contrast, some unknown virus sequences identified in the study exhibited low prevalence in the bee populations or were found to occur in localized instances. For example, bee dicistrovirus BD 4 was detected solely in *A. dorsata*, with a viral count of 21.64 RPM (Table S2). Other rare viruses, such as bee dicistrovirus BD 8 and bee dicistrovirus BD 9, were detected in *A. dorsata* and *A. florea*, respectively. Figure 4 reveals several interesting patterns of co-occurrence. Lake Sinai virus (LSV) occurrence was evident solely in *A. mellifera* libraries in the presence of BQCV (from Mymensingh-Shutiakhali, Dhaka, Shatkhira, and Tangail). There is a statistically significant dependence between the presence of BQCV and the occurrence of LSV (Fisher's exact test, *P*-value = 0.02778). Bee-associated cripavirus 1, bee-associated dicistrovirus 1, and *Planococcus ficus*-associated dicistrovirus 1 appeared to co-occur in

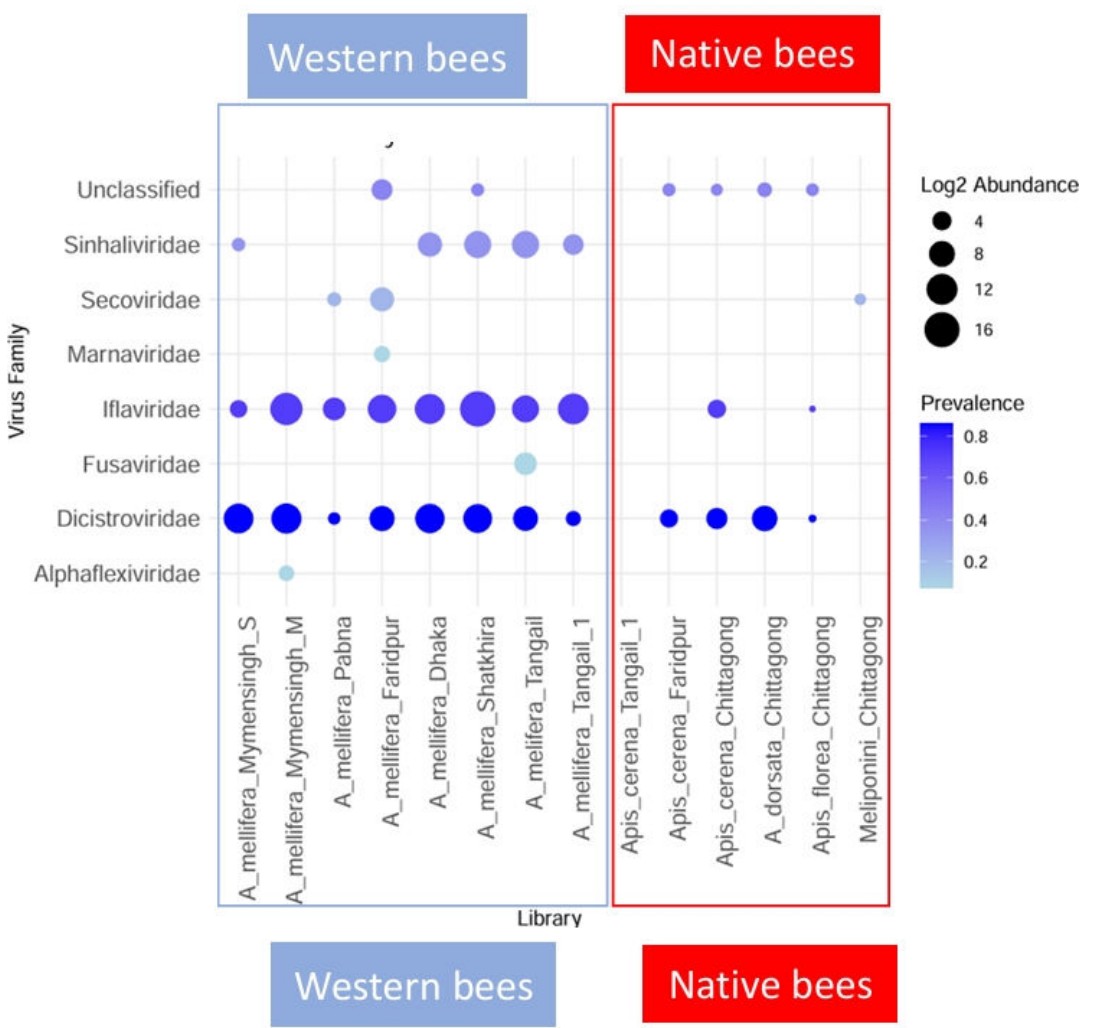

FIG 2 Prevalence and abundance of viral families in bee species in Bangladesh. The x-axis of the bubble plot shows the libraries, while the y-axis shows the virus families. The size of each bubble represents the log2-transformed abundance of the virus family in each library, while the color gradient indicates the prevalence across libraries. Larger bubbles indicate higher abundance, and darker blue colors indicate higher prevalence.

the bee libraries, particularly in cases where BQCV was absent or present in minimal amounts (Fig. 4). A single overall Fisher's exact test indicated a statistically significant negative association between the presence of BQCV and the detection of bee-associated cripavirus 1, bee-associated dicistrovirus 1, and *Planococcus ficus*-associated dicistrovirus 1 ($P = 0.02857$; odds ratio = 0; 95% CI: 0.000–1.080). This suggests that these viruses are unlikely to co-occur with BQCV, potentially due to competitive exclusion. However, because the confidence interval includes 1, this relationship should be interpreted with caution, and additional data would help confirm the trend. As this was a single test rather than multiple comparisons, statistical correction for multiple testing was not required.

## Diversity of insect viruses

The viral richness investigation showed that both native and western bees generally harbor several viral species, ranging from three to nine species in the native species, compared to two to eight in western bees (*A. mellifera*) (Fig. 5A; Table S3), with no difference in species richness (W = 10, *P*-value = 0.8726). The populations also showed a range of alpha diversities, with significantly higher diversity in native species compared to *A. mellifera*, as indicated by both the Shannon index (W = 28, *P*-value = 0.048) and the

A. *Iflaviridae* family

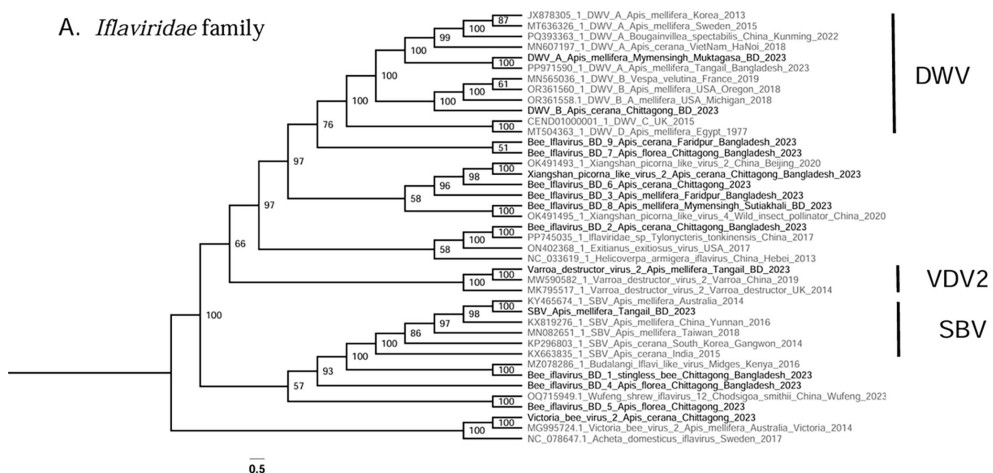

B. *Dicistroviridae* family

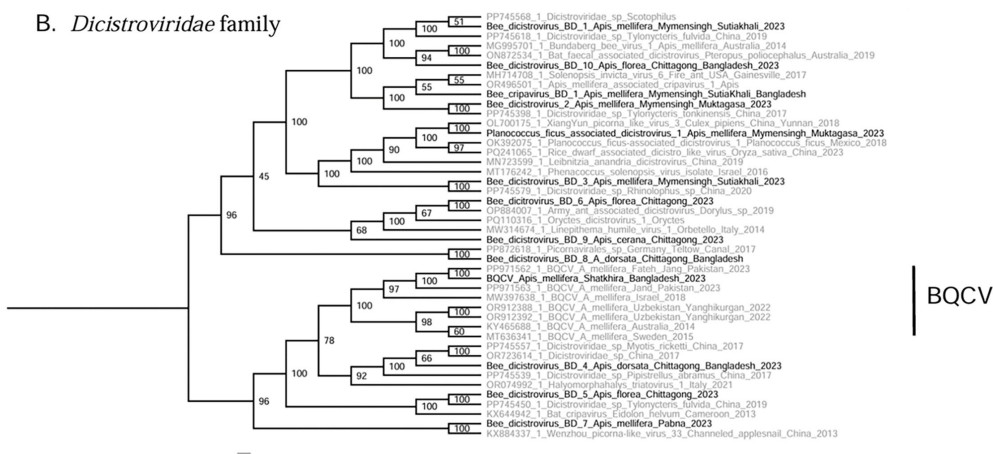

C. *Sinhaliviridae* family

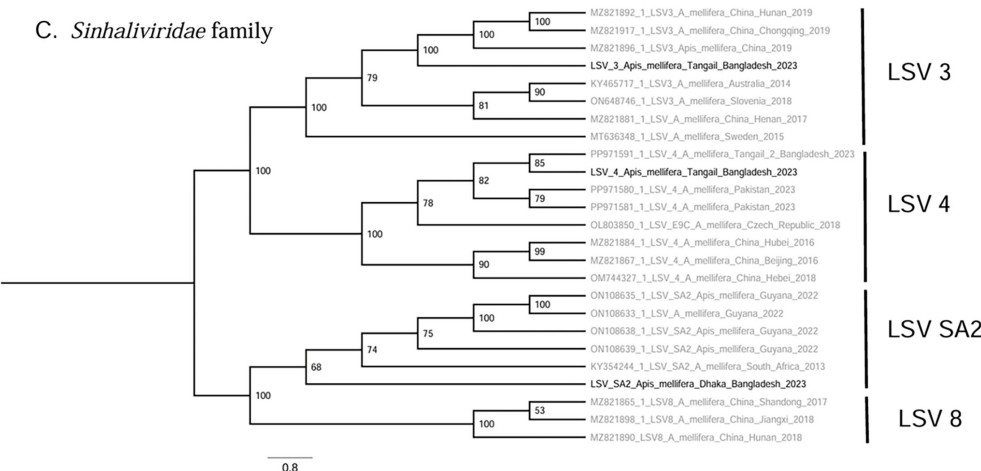

**FIG 3** Phylogenetic analyses of insect viruses in bee species in Bangladesh. (A) The phylogenetic clustering of viruses from the Iflaviridae family (15 virus OTUs), which highlights distinct lineages and evolutionary relationships among known viruses, such as deformed wing virus (DWV) and sacbrood virus (SBV). (B) The evolutionary relationships of viruses from the Dicistroviridae (Continued on next page)

**Fig 3 (Continued)**

family (13 virus OTUs), including the black queen cell virus (BQCV). (C) The Sinhaliviridae family (three virus OTUs) depicting phylogenetic clusters for the Lake Sinai virus (LSV) strains. Phylogenetic trees were constructed using RAxML with 1,000 bootstrap iterations, and bootstrap values are indicated for branch support. Sequences were aligned using MAFFT, and the GTR + G model was used to predict relationships.

Simpson index (W = 28, *P*-value = 0.048) (Fig. 5B and C; Table S3). A PERMANOVA analysis of Bray-Curtis distances, with host species as a factor, reveals a significant difference in the virome composition between western honey bees and native honey bees (F = 1.7914, *P* = 0.019, d.f. = 1). This clear distinction between the viromes of western honey bees and native honey bees is also found in a principal coordinates analysis (PCoA) of the Bray-Curtis distances (Fig. 5D), with native bees closely clustering together, while the western bees are widely spaced.

## Plant viruses

Ten potential viruses related to plants or fungi were identified from bee libraries through sequence analyses followed by phylogenetic analysis (Fig. 6A). Each virus appeared in only one library (Fig. 6B). Secoviruses, such as tomato black ring virus and rehmannia torradovirus, were identified in *A. mellifera*, and a novel virus sequence, bee-associated nepovirus BD1, was detected in *Trigona* sp. They clustered with secoviruses previously reported in different plant species from other countries in the phylogenetic tree (Fig. 6A). Similarly, potexviruses such as bee-associated potexvirus BD 1 and bee-associated potexvirus BD 2, potyviruses such as chili veinal mottle virus BD, novel bee-associated plant RNA virus 1, and marnaviruses such as Lactuca sativa marnavirus and *Marnaviridae* sp., identified in *A. mellifera* from different locations in Bangladesh, cluster with other virus sequences previously reported in plant species. We also detected a mycovirus sequence, tentatively named bee-associated *Fusaviridae* sp., from an *A. mellifera* library in Tangail. Interestingly, none of the tested native *Apis* species were positive for plant or fungal viruses.

## DISCUSSION

Using poly(A)-selected high-throughput RNA sequencing (RNA-seq), we explored the viromes of the non-native, managed western honey bee (*Apis mellifera*) and native bee species in Bangladesh, including *A. cerana*, *A. dorsata*, *A. florea*, stingless bees (*Trigona* spp.), and the inquiline species of the western honey bee, *Aethina tumida*. Although our data set comprises a limited number of samples (*n* = 15) collected from distinct geographic regions and diverse hosts, we employed rigorous quality control measures and conservative assembly strategies to address these challenges. This study provides valuable insights into virus occurrence and diversity across multiple bee taxa and lays the groundwork for future investigations into interspecific viral transmission and potential spillover risks in a biodiversity-rich region. We identified 50 distinct viral OTUs, including known insect viruses, novel insect viruses, and plant viruses. The western honey bee populations showed a distinct virome, with a higher proportion of viral reads but less alpha diversity than the native bees, which also showed more variation in beta diversity (Table 1; Fig. 5). This shows that *Apis mellifera* is a potential source for viruses in Bangladesh, but that native bees also carry diverse viromes.

This work confirms that, while diverse virus families infect honey bees (21), the Iflaviridae and Dicistroviridae are the most common and dominant viral families (15, 21, 22). However, as we used poly(A)-selected RNA sequencing, this pattern may reflect not only true biological abundance but also methodological bias, since the approach enriches for polyadenylated viruses while underrepresenting others. The family Iflaviridae includes viruses with major threats to global bee populations, such as DWV and SBV. DWV, transmitted by the *Varroa destructor* mite, causes wing deformities and, in conjunction with *Varroa*, causes high overwinter colony mortality (23). In our study,

DWV-A was detected in all examined libraries of *A. melifera*, where four libraries also showed the presence of DWV-B (Fig. 4). DWV-B has been shown to have higher virulence at the colony level in *A. mellifera* in Europe (23). A shift from DWV-A to DWV-B was first observed in *A. melifera* in Europe, with more recent detections in Asia (24–26). While DWV-B appears to be spreading in South Asia, these results are in line with DWV-A still remaining more prevalent in Asia (10). While SBV, on the other hand, is comparatively rare and of little concern in western honey bees, it has led to severe colony losses in outbreaks of *A. cerana* across Asia (9). It can cause pupation failure and death in larvae

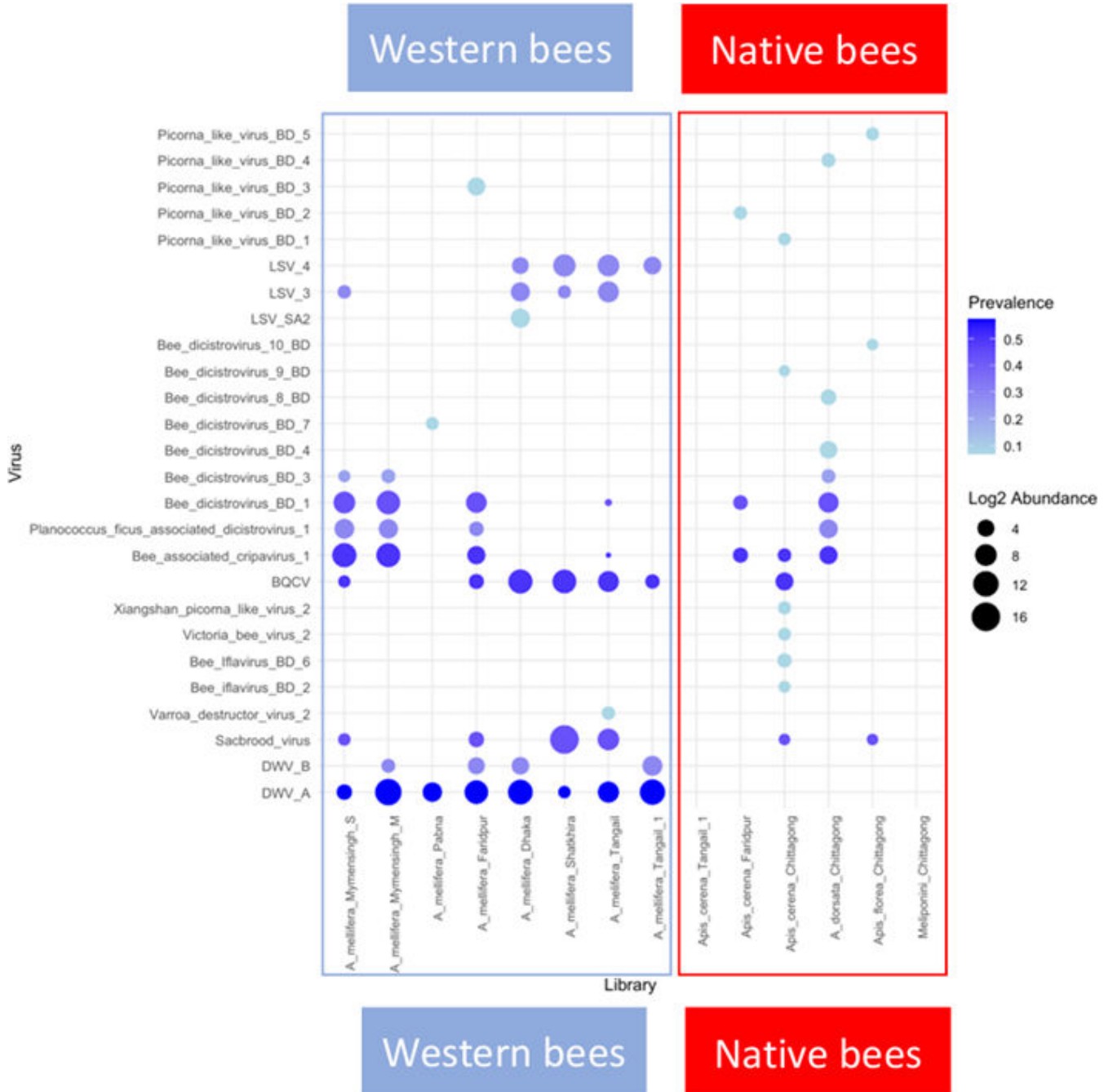

**FIG 4** Prevalence and abundance of insect viruses in bee species in Bangladesh. The x-axis of the bubble plot shows the libraries, while the y-axis shows the insect virus species. The size of each bubble represents the normalized abundance of the virus family in each library, while the color gradient indicates the prevalence across libraries. Larger bubbles indicate higher abundance, and darker blue colors indicate higher prevalence.

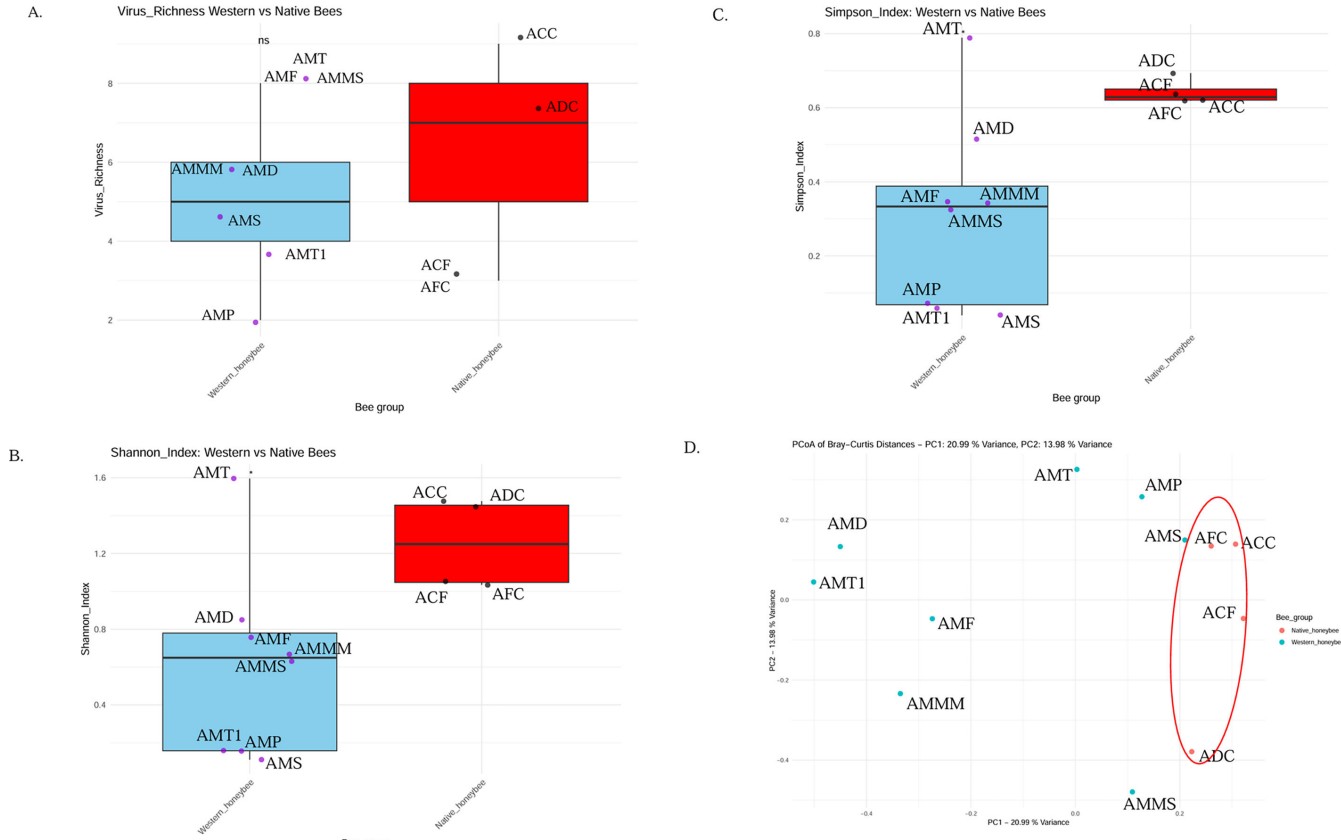

**FIG 5** Diversity of insect viruses in western and native bee species. (A) Richness of insect virus species in western honey bee and native honey bee species (B) Shannon diversity index comparison between western and native honey bees. (C) Simpson diversity index comparison between western and native honey bees (D) Principal Coordinate Analysis (PCoA) based on Bray-Curtis distance of bee viromes. Legends: ACF- A_cerena_Faridpur, ACC-A_cerana_Chittagong, ADC-A_dorsata_Chittagong, AFC-A_florea_Chittagong, AMMS-A_mellifera_Mymensingh_S, AMMM-A_mellifera_Mymensingh_M, AMP-A_mellifera_Pabna, AMF-A_mellifera_Faridpur, AMD-A_mellifera_Dhaka, AMS-A_mellifera_Satkhira, AMT-A_mellifera_Tangail, and AMT1-A_mellifera_Tangail_1.

and adults (27). This globally distributed virus was found in *A. mellifera*, *A. cerana*, and *A. florea* in Bangladesh (Fig. 4). Variation in virus genotypes in different bee hosts has been reported (15, 28–30), and the high virulence in *A. cerana* raises concerns about possible spillovers and recombination of viral variants, as seen in DWV. These findings emphasize the need for continuous surveillance and research on the impact of both DWV and SBV on honey bee populations in Asia, considering their prevalence, genetic diversity, and spread.

The Dicistroviridae include important viruses, such as acute bee paralysis virus (ABPV), BQCV, Kashmir bee virus (KBV), and Israeli acute paralysis virus (IAPV) (15, 22). Among these, IAPV and BQCV were reported as commonly occurring in honey bee colonies in different countries (22). While we did not detect ABPV, KBV, and IAPV in our studied samples from Bangladesh (Fig. 4), BQCV was detected in seven *A. mellifera* libraries and one *A. cerana* library in our study. Other dicistroviruses, i.e., bee-associated cripavirus 1, bee-associated dicistrovirus 1, and *Planococcus ficus*-associated dicistrovirus 1, were found to co-occur in the *A. mellifera* and *A. dorsata* libraries, particularly in cases where BQCV was absent or present in lower amounts. A reminiscent pattern, with the presence of BQCV seemingly inversely related to the presence of other Dicistroviridae such as ABPV, IAPV, KBV, and SBPV, has also been reported in other studies of honey bees (31, 32). Here, we find a further potential pattern of co-occurrence with BQCV: LSV. This co-occurrence was only evident in *A. mellifera* libraries (Fig. 4). LSV could infect

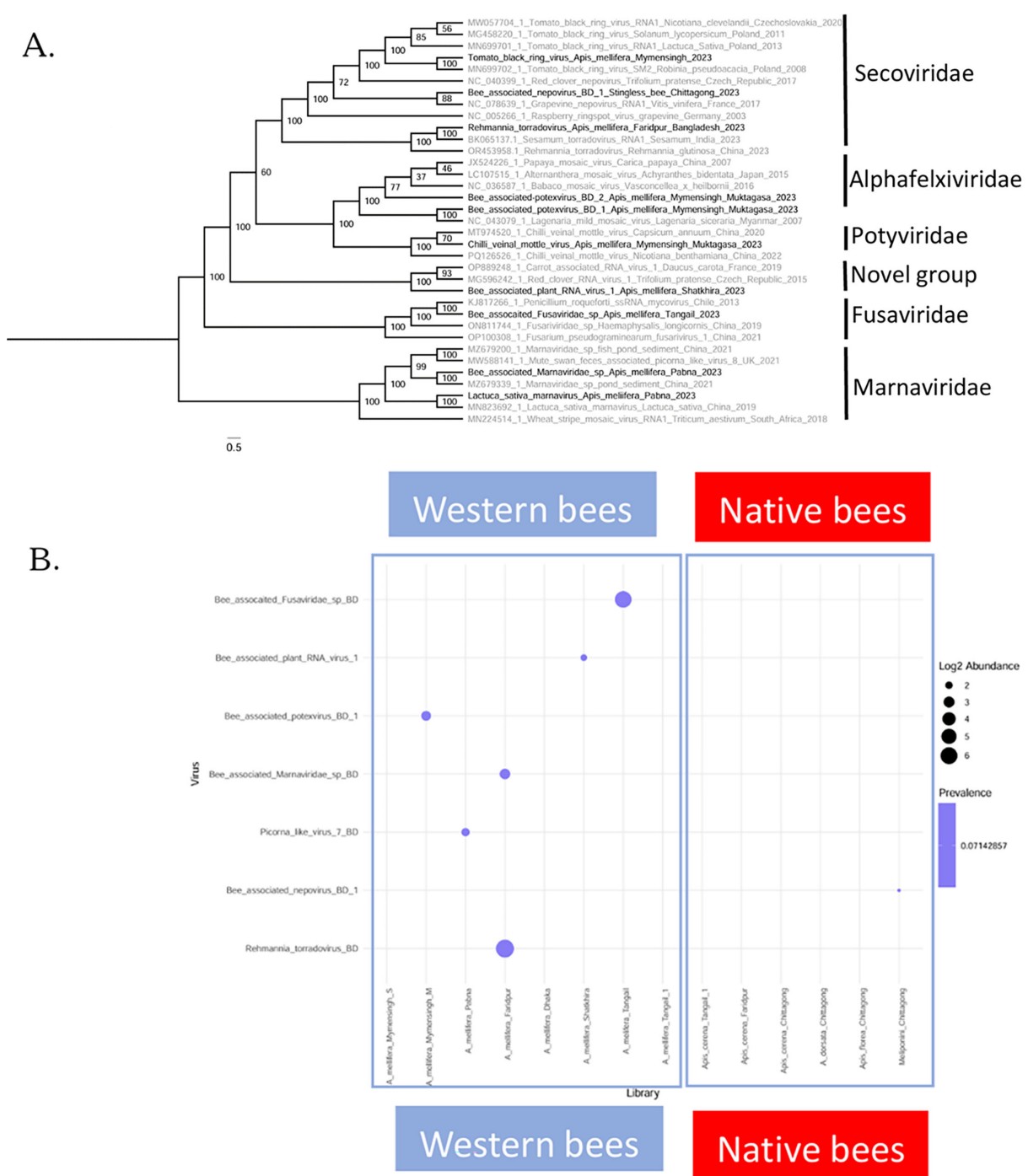

**FIG 6** Phylogenetics and distribution of plant-related viruses in bee species in Bangladesh. (A) Phylogenetic analysis of plant-related viruses identified in this study. Phylogenetic trees were constructed using RAxML with 1,000 bootstrap iterations, and bootstrap values are indicated for branch support. Sequences were aligned using MAFFT, and the GTR + G model was used to predict relationships. (B) Bubble plot showing the prevalence and abundance of plant-related viruses in bee species.

ants, solitary bees, bumble bees, and hornets besides honey bees, indicating possible cross-species transmission (33, 34). Such potential patterns of exclusion or co-infection warrant further monitoring across populations and potential host species, as well as experimental verification.

In addition to these well-known viruses, several novel Dicistroviridae (bee dicistrovirus BD 1 to 10), Iflaviridae (bee iflavirus BD 1 to 9), and unclassified viral OTUs were found

in *A. mellifera*, *A.cerana*, *A. dorsata*, and *A. florea*. As the virome of more populations and species of bees is investigated by *de novo* sequencing methods, the number of viral OTUs has been increasing (21). While these viruses may have a limited geographic or host distribution and their degree of virulence is unknown, it is important to obtain a fuller picture of the virome of honey bees, particularly given the potential increasing need for pollinators to maintain global food security and the linked increase in the potential for transmission across species and geographic regions. For example, a novel dicistrovirus identified in *A. mellifera* from the Netherlands was later identified in *A. florea* from India (22, 35). Similarly, rare viruses may be rare precisely because they have high virulence. Understanding these less common viruses and their interactions with other honey bee viruses, potential vectors, and biotic or abiotic stressors is thus crucial for future research into pathogens of these beneficial insects.

This study revealed stark differences in the composition of viromes of *A. mellifera* (western honey bees) and native bees in Bangladesh. The well-known and globally distributed viruses DWV-A and DWV-B, SBV, LSV, and BQCV were not detected or showed very low viral reads in bees native to Bangladesh in our study compared to *A. mellifera* (Fig. 4). Generally, the virome of *A. mellifera* has been found to be quite distinct from more distantly related Apidae (e.g., bumble bees and stingless bees, solitary bees) (9, 21, 36, 37). However, an RNAseq study including global *A. mellifera* and *A. cerana*, *A. dorsata,* and *A. florea* from India did not detect such clear differentiation, with all Asian honey bee species also positive for DWV (22). Additionally, targeted qPCR-based surveys of *A. mellifera* and *A. cerana* in China have shown the presence of typical honey bee viruses (DWV, BQCV, IAPV, SBV), albeit to some extent at lower prevalence or titer in *A. cerana*, except for SBV, which showed the opposite pattern. However, it should be noted that we applied stringent thresholds, including a 15% minimum genome coverage threshold for the reference viral sequence, to minimize false positives in viral read estimation in this study. Here, we found lower alpha diversity in western honey bees compared to native honey bees, while viral richness did not differ between these groups (Fig. 5). This can be explained by the viromes of western honey bees being dominated by few viral strains, particularly DWV-A and DWV-B, in the presence of the *Varroa* mite. Native honeybees, in contrast, show similarly rich, but more evenly distributed, sets of viruses. However, it should be noted that, with the exception of *A. cerana* from Faridpur and Tangail (SRR2840858), all of the other native bees were collected in Chittagong division, from which we could not obtain *A. mellifera*. Thus, the difference between western honey bees and native honey bees, as well as the difference to studies from India (22) and China (9), could at least partially be due to environmental factors, such as climate or agricultural practices. These results emphasize the importance of understanding the potential impact of viruses on honey bees as well as the potential for cross-species transmission in different agricultural and climatic environments in South Asia, where commercial pollination and apiculture are developing at a fast pace and are crucial for food security and poverty alleviation.

In addition to diseases that can harm pollinators, bee species can also carry plant pathogens (22, 37, 38). Indeed, we also detected several plant viruses (e.g., the Secoviridae tomato black ring virus, bee-associated nepovirus, and Rehmannia torradovirus), including novel OTUs (bee-associated plant RNA virus 1, bee-associated nepovirus, and bee-associated potexvirus BD 1) (see Fig. 6). In contrast to the typical bee viruses, whose distribution was found to be determined by phylogenetic relationships between honey bees and bumble bees in a European study, the plant virome carried by pollinators was found to be structured by season and pollinator niche overlap (21). Similarly, Kadlečková et al. (39) found that plant viruses, but not bee viruses, showed small-scale geographic clustering. This shows that the plant virome carried by bees is strongly affected by the plants from which bees forage at any given time and place. It is clear that bees can vector plant diseases via pollination (40, 41). Whether bees can also be active vectors of plant viruses remains unclear, although replication of tobacco ringspot virus in honey bees has been reported (38). At the same time, bee pollination can also reduce the

vertical spread of diseases in self-pollinated plants (42). From an applied point of view, sequencing pollen collected by bees can be used to monitor the prevalence and spread of plant diseases (43). The presence of plant pathogens in pollinators such as honey bees can also give fundamental insights into plant-pollinator networks, particularly in difficult-to-observe tropical ecosystems such as those in Southeast Asia, where insect pollinators frequently feed in the canopy of flowering trees. Studying the association between plant viruses and bees has great potential in Asia, with the opportunities to better understand insect pollination in tropical agriculture and natural environments, as well as enabling monitoring for plant diseases over large areas. Furthermore, the potential for plant viruses to replicate in bees warrants further research. While cross-king-dom replication is evident in plant viruses such as rice stripe virus, vectored by small brown planthopper (*Laodelphax striatellus*) (44), the potential for such active vectoring remains to be tested in bees, particularly in tropical ecosystems.

Southeast Asia is the center of honey bee biodiversity, but has also given rise to several global emerging bee parasites and pathogens (13, 45). Here, we have shown that non-native western honey bees have a distinct virome from native honey bees in Bangladesh, characterized by high viral reads and viruses pathogenic to *A. mellifera* and *A. cerana*. This study thus highlights the need for comprehensive virus surveillance programs in Bangladesh and throughout Southeast Asia, including both known and novel viruses carried by bees. This research also calls for future studies on how these viruses impact wild and managed bee species. Moreover, monitoring the viruses across different wild bee populations in agricultural and wild ecosystems, and regions will be crucial for understanding the fundamental ecology of pathogens infecting wild bees, preventing spillovers from wild to managed bees, and reducing the potential for new emerging bee diseases.

## MATERIALS AND METHODS

### Sample collection and processing

The study analyzed samples of diverse foraging bee species collected opportunistically across multiple regions in Bangladesh, providing an overview of occurrence of viruses in different bee populations (Fig. 1; Table 1). Sampling was uneven across species and sites, with collections focused on single foraging locations per region due to field and logistical constraints. Therefore, this study is exploratory in nature. Samples of *Apis mellifera* (western honey bee) were collected from single foraging sites in Pabna, Faridpur, Dhaka, Satkhira, and Tangail and from two different sites in Mymensingh. *A. cerana* (Asian honey bee) was sampled from single sites in Faridpur and Chittagong. In Chittagong, we were also able to sample *A. dorsata* (giant honey bee), *A. florea*, and stingless bees identified as *Trigona* sp., which may include multiple species due to taxonomic uncertainty. Additionally, the small hive beetle (*Aethina tumida*), a known parasite of *A. mellifera*, was collected from a hive in Dhaka. After collection, each sample was immediately preserved in RNAlater solution to ensure RNA integrity, followed by RNA extraction and sequencing for virome analysis (46). Total RNA was extracted from tissue of laterally single-bisected bee (half-bee) following previously described methods (18, 19). The tissue was homogenized at 5 m/s for 25 sec in three cycles with 20-second pauses using a FisherBrand Bead Mill 24. RNA extraction was performed using 1.3 mL TRI Reagent and 0.1 mL 1-bromo-3-chloropropane (both from Sigma-Aldrich). RNA was eluted in 80 μL of nuclease-free water. RNA concentrations were quantified using the QuantiFluor RNA System (Promega). RNA samples were then pooled by species and location, with each pool containing RNA from five individuals of the same species, generating 13 RNA libraries for polyA-selected RNASeq on the DNBSEQ platform (BGI Bioinformatics, China) with 150 bp paired-end reads, yielding sufficient data for viral analysis. Additionally, two publicly available RNA-seq data sets from *A. mellifera* (SRR28408579) and *A. cerana* (SRR2840858) from Tangail, Bangladesh, were included, bringing the total to 15 data sets used in the viral detection and diversity analysis.

## Bioinformatic processing of RNA-Seq data

Raw sequencing data were processed using SOAPnuke software to ensure high-quality data for downstream analysis (47). Reads with ≥25% adapter sequence match (allowing up to two base mismatches) were removed to eliminate adapter contamination. Additionally, reads shorter than 150 base pairs, with ≥0.1% unknown bases (N), and reads containing polyX stretches longer than 50 base pairs were also removed. Reads were additionally quality filtered by removing those where more than 40% of bases had a Phred quality score <20, and the Phred + 33 quality score system was used to assess the accuracy of base calls. These rigorous quality control steps ensured the retention of high--quality data for further analysis. The sequencing data quality was further assessed using FastQC, which identified concerns such as low per-base quality, GC content anomalies, and adapter contamination (https://www.bioinformatics.babraham.ac.uk/projects/fastqc/). Low-quality reads and adapter sequences (if present) were removed using Sickle. To remove host genetic material, quality-filtered reads were aligned using Bowtie2 to the respective host genomes: *A. mellifera*, *A. cerana*, *A. florea*, *A. dorsata*, and *Aethina tumida* (48). This step reduced the volume of non-viral RNA-seq data to decrease data complexity and improve virus detection during *de novo* assembly (Trinity), particularly by facilitating contig reconstruction for subsequent blast analysis. However, for downstream analyses such as viral read counts (used as a proxy for viral load), we used the entire set of quality-filtered RNA-seq reads and mapped them directly to a reference virus database. For *Trigona* sp., RNA-seq data were processed without host read removal, using the quality-filtered reads directly for downstream analysis.

## Identification of viral OTUs

Non-host reads were assembled using Trinity (49), with a minimum length of 500 nucleotides. However, for *Trigona* sp., all quality-filtered reads were used for assembly, which may have included host-derived sequences. After *de novo* assembly, BLASTX was used to identify potential viral sequences by comparing the assembled contigs to the NCBI virus protein database (https://www.ncbi.nlm.nih.gov/labs/virus/vssi/#/) (49). Viral sequences were classified taxonomically according to their top BLASTx hits, identifying viral families and lineages. To ensure the accuracy of these identifications, we manually verified the closest relatives against the GenBank database, cross-referencing the results to resolve potential misclassifications or sequence match discrepancies. We focused on sequences containing the RNA-dependent RNA polymerase (RdRp) domain for all downstream analyses, including virus detection, diversity assessment, and phylogenetic reconstruction, which serves as a key diagnostic for viral sequences (50), as it is essential for RNA virus genome replication and is therefore highly conserved. The virus sequences were studied by identifying open reading frames (ORFs) using NCBI's ORF Finder and translating them into amino acid sequences. The presence of RdRp domains was confirmed by searching NCBI's Conserved Domain Database and Pfam for specific RdRp-related domains to validate domain identity and function. For our study, RdRp-containing viral sequences were classified into operational taxonomic units (OTUs) based on a 90% amino acid sequence identity threshold. This classification provided information about the composition of the viral community and enabled the identification of novel variants within different viral populations.

Phylogenetic relationships were investigated using maximum likelihood phylogenetic trees. These trees were constructed with 1,000 bootstrap iterations using RaxML (Randomized Axelerated Maximum Likelihood) (51) using the GTR + G (General Time Reversible with Gamma Distribution) sequence evolution model. Sequence alignments were constructed using MAFFT (52). The obtained maximum likelihood trees were visualized using FigTree.

## Estimation of viral reads and normalization by RPM and genome length

To estimate viral reads, clean reads from RNA sequencing were aligned to assembled viral contigs using Bowtie2 with the alignment mode set to highly sensitive (48). The amount of reads mapping to each viral OTU was normalized by reads per million (RPM), giving the number of viral reads scaled to a million total reads in the cleaned library. Furthermore, the genome length of each virus was included in the normalization procedure, ensuring that viral read estimates were adjusted not only for total sequencing depth and library size, but also for viral genome size. For distribution and diversity analyses, we set a 15% minimum genome coverage threshold for the reference viral sequence to minimize false positives in viral read estimation, ensuring that only reliable viral reads with adequate coverage are included in the analysis. This helped eliminate low-abundance contaminants and potential errors that could lead to false matches. However, this stringent cut-off led to no viral reads being recorded for two libraries: *Trigona* sp. and *A. cerana* from Tangail (SRR2840858). This outcome may reflect true biological absence of detectable viruses at the time of sampling or viral loads below the detection threshold imposed by our 15% genome coverage filter. Moreover, given our use of poly-A selected RNA, we may miss viral genomes that lack polyadenylation.

## Analysis of virome composition and diversity

All analyses were performed in R (https://www.r-project.org/), using the ggplot2 package for plotting (53), dplyr for data handling (https://github.com/tidyverse/dplyr), and scales for scaling log-transformed data (https://github.com/r-lib/scales), for example, for prevalence and abundance based on viral reads. We used species richness as well as the Shannon and Simpson diversity indices to assess virome alpha diversity using the vegan package in R (https://CRAN.R-project.org/package=vegan). Beta diversity was investigated via Bray-Curtis distances calculated using the vegan package. PERMANOVA was then applied to look for significant changes across bee groups over 999 permutations using the adonis() function (54). For visualization, the two first coordinates of a Principal Coordinate Analysis (PCoA) were plotted. The two libraries, *Trigona* sp. and *A. cerana* from Tangail (SRR2840858), which did not contain viral reads based on our conservative cut-off (see above), were excluded from these analyses.

## ACKNOWLEDGMENTS

We thank S.M. Mainul Anwar (Alwan Honey, Bangladesh), Md. Abul Kalam (Ayesha Honey, Bangladesh), Gazi Sakib Ahmad (Maag Honey, Bangladesh), Raduanul Hasan (Mymensingh, Bangladesh), Md. Aminul Islam (Maya Agro, Mymensingh, Bangladesh), Elmur Reza (Bangladesh Sugarcrop Research Institute), Sajedur Rahman Sajib (Bangladesh Sugarcrop Research Institute), and Tilak Ghosh (Department of Agricultural Extension, Bangladesh) for their help during sample collection. At Ulm University (Germany), we thank Dr. Vincent Doublet, Dr. Jana Dobelmann, Sofia Palacios Trujillo, Dr. Gerd Mayer, Manuel Stech Domene, and Dr. Fanni Borveto for their help during molecular work and bioinformatic analyses.

The authors acknowledge support by the High Performance and Cloud Computing Group at the Zentrum für Datenverarbeitung of the University of Tübingen, the state of Baden-Württemberg through bwHPC and the German Research Foundation (DFG) through grant no. INST 37/935-1 FUGG. Islam Hamim was funded through a Georg Forster fellowship (Ref 3.5 - BGD - 1217640 - GF-P) from the Alexander von Humboldt Foundation, Germany.

## AUTHOR AFFILIATIONS

[1]Department of Plant Pathology, Bangladesh Agricultural University, Mymensingh, Bangladesh

²Institute of Evolutionary Ecology and Conservation Genomics, University of Ulm, Ulm, Germany

## AUTHOR ORCIDs

Islam Hamim ⓘ http://orcid.org/0000-0002-0200-8108

## AUTHOR CONTRIBUTIONS

Islam Hamim, Conceptualization, Data curation, Formal analysis, Funding acquisition, Investigation, Methodology, Project administration, Resources, Software, Supervision, Validation, Visualization, Writing – original draft, Writing – review and editing | Lena Wilfert, Conceptualization, Investigation, Methodology, Project administration, Supervision, Writing – review and editing

## DATA AVAILABILITY

The data sets supporting this article have been uploaded as part of the electronic supplementary material. RNA-seq reads from this study have been deposited to the NCBI Sequence Read Archive under SRA project accession no. PRJNA1241539. In addition, known and novel virus sequences identified in this study have been provided as supplemental material. Additional data required by readers will be available upon request.

## ADDITIONAL FILES

The following material is available online.

### Supplemental Material

**Supplemental material S1 (Spectrum01971-25-s0001.csv).** Pairwise sequence similarities between DWV and potential outgroups of bee species in Bangladesh.
**Supplemental material S2 (Spectrum01971-25-s0002.txt).** Sequences of known and novel viruses.
**Supplemental figures (Spectrum01971-25-s0003.docx).** Figures S1 to S3.
**Supplemental tables (Spectrum01971-25-s0004.docx).** Tables S1 to S3.

### Open Peer Review

**PEER REVIEW HISTORY (review-history.pdf).** An accounting of the reviewer comments and feedback.

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
