## [Reviewer comments · Microbiology Spectrum]

Microbiology Spectrum

Insights into the Viral Landscape of the Western Honey Bee and Native Bees in Bangladesh

Islam Hamim and Lena Wilfert

Corresponding Author(s): Islam Hamim, Bangladesh Agricultural University

Review Timeline:

Submission Date:	June 27, 2025
Editorial Decision:	September 1, 2025
Revision Received:	September 15, 2025
Accepted:	September 17, 2025

Editor: Jonathan Snow

Reviewer(s): Disclosure of reviewer identity is with reference to reviewer comments included in decision letter(s). The following individuals involved in review of your submission have agreed to reveal their identity: Ivan Toplak (Reviewer #1); Dominika Kadleckov (Reviewer #2)

Transaction Report:

DOI: <https://doi.org/10.1128/spectrum.01971-25>

Re: Spectrum01971-25 (**Insights into the Viral Landscape of the Western Honey Bee and Native Bees in Bangladesh**)

Dear Dr. Islam Hamim:

Thank you for the privilege of reviewing your work. Below you will find my comments, instructions from the Spectrum editorial office, and the reviewer comments. As you will see, the reviewers were quite positive, but requested a number of revisions to make the manuscript suitable for publication. Please make sure to address all of the reviewer concerns including these points in any resubmission.

Revision Guidelines

Sincerely,
Jonathan Snow
Editor
Microbiology Spectrum

Reviewer #1 (Comments for the Author):

This manuscript is very well written and suitable for publication.
I have only few minor remarks, which need to be implemented before publication.

1. page 6, line 136-add which high-throughput technology was used for sequencing

2. Page 7, line 162: RPM-first mentioned, explain
3. Page 12, line 263: (n=13), check if this is correct, in abstract you have n=15
4. Page 17, line 405. Add how many samples (*Apis mellifera*) were tested. Add also how many bees were included into one sample.
5. Page 19, line 445-check one space in bracket
6. Page 20, line 461-469, check some words are in bold

Reviewer #2 (Comments for the Author):

This manuscript provides valuable descriptive insights into the virome of both managed and native bees in Bangladesh using RNA sequencing. The topic is timely and relevant, and the work contributes to the growing body of regional pollinator virome studies. The manuscript is generally well written and clear. I have a few comments for the authors to consider:

1. Bias in methodology

The study relies on poly(A)-selected RNA sequencing. This inherently enriches for polyadenylated RNA viruses and excludes some viral groups. Consequently, statements such as "Using high-throughput RNA sequencing, we show that viruses of the order Picornavirales dominate the viral landscape of both western and native bees" (lines 24-26) should be moderated, as this conclusion may reflect methodological bias as much as true biological dominance. Please consider rephrasing these claims to more accurately reflect the scope of the data.

2. Novel virus genomes and data availability

While raw RNA-seq reads have been deposited, several new or unreported viral genomes are described in the manuscript (e.g., novel Iflaviridae and Dicistroviridae). If the viruses are near-complete it might be beneficial to describe them further and deposit them in GenBank and provide the accession in Data availability section.

3. Choice of assembler

Trinity is a standard tool for transcriptome assembly and is used in virome studies, but viruses are not transcripts, and metagenomic assemblers (e.g., MEGAHIT, metaSPAdes) may perform better in reconstructing viral genomes, particularly in mixed samples. While Trinity is acceptable here in combination with stringent QC, it may be useful to consider alternative assembly tools in future studies.

This is a well-executed descriptive study that will add to the global picture of bee viromes. The manuscript would benefit from tempering some claims, clarifying data deposition, and more explicitly acknowledging methodological constraints (poly-A bias, sampling unevenness).

This manuscript provides valuable descriptive insights into the virome of both managed and native bees in Bangladesh using RNA sequencing. The topic is timely and relevant, and the work contributes to the growing body of regional pollinator virome studies. The manuscript is generally well written and clear. I have a few comments for the authors to consider:

1. Bias in methodology

The study relies on poly(A)-selected RNA sequencing. This inherently enriches for polyadenylated RNA viruses and excludes some viral groups. Consequently, statements such as “*Using high-throughput RNA sequencing, we show that viruses of the order Picornavirales dominate the viral landscape of both western and native bees*” (lines 24–26) should be moderated, as this conclusion may reflect methodological bias as much as true biological dominance. Please consider rephrasing these claims to more accurately reflect the scope of the data.

2. Novel virus genomes and data availability

While raw RNA-seq reads have been deposited, several new or unreported viral genomes are described in the manuscript (e.g., novel Iflaviridae and Dicistroviridae). If the viruses are near-complete it might be beneficial to describe them further and deposit them in GenBank and provide the accession in Data availability section.

3. Choice of assembler

Trinity is a standard tool for transcriptome assembly and is used in virome studies, but viruses are not transcripts, and metagenomic assemblers (e.g., MEGAHIT, metaSPAdes) may perform better in reconstructing viral genomes, particularly in mixed samples. While Trinity is acceptable here in combination with stringent QC, it may be useful to consider alternative assembly tools in future studies.

This is a well-executed descriptive study that will add to the global picture of bee viromes. The manuscript would benefit from tempering some claims, clarifying data deposition, and more explicitly acknowledging methodological constraints (poly-A bias, sampling unevenness).

Dear Editor,

We would like to sincerely thank you and the reviewers for your time, effort, and thoughtful feedback and suggestions on our manuscript (Spectrum01971-25, *Insights into the Viral Landscape of the Western Honey Bee and Native Bees in Bangladesh*). This study presents the first comprehensive look at bee viruses in Bangladesh, a country rich in biodiversity. In Bangladesh, pollinators play a vital role in agriculture and livelihoods, yet research investigating viruses in bee pollinators remains underexplored. We have carefully revised the manuscript in line with the reviewers' suggestions, including rephrasing and clarifying our claims more accurately reflect the scope of the data, improving the data availability statement, and clearly acknowledging methodological limitations. We believe that these revisions have strengthened and improved the manuscript and brought it closer to publication in the Microbiology Spectrum. We have uploaded both a clean revised version and a marked-up copy, and provide our detailed point-by-point response below.

Reviewer #1 (Comments for the Author):

This manuscript is very well written and suitable for publication. I have only few minor remarks, which need to be implemented before publication.

reviewer comment--1. page 6, line 136-add which high-throughput technology was used for sequencing

Author response:

We have added that high-throughput, poly(A)-selected RNA sequencing (RNA-Seq) on the DNBSEQ platform (BGI, China) was used for sequencing. The sentence has been revised to read:

“In this study, we used high-throughput, poly(A)-selected RNA sequencing (RNA-Seq) on the DNBSEQ platform (BGI, China) to investigate the viral landscape across different honey bee species in Bangladesh.”

2. Page 7, line 162: RPM-first mentioned, explain

Author response:

We have explained as per reviewer 1's suggestion. The sentence has been revised to read:

"Iflaviridae was found as the most abundant family, with 95098.55 reads per million (RPM), followed by dicistroviridae (7718.257 RPM)."

3. Page 12, line 263: (n=13), check if this is correct, in abstract you have n=15

Author response:

Thank you for spotting this, n=15 is correct. The sentence has been corrected and revised to read: " Although our dataset comprises a limited number of samples (n = 15) collected from distinct geographic regions and diverse hosts, we employed rigorous quality control measures and conservative assembly strategies to address these challenges. "

4. Page 17, line 405. Add how many samples (*Apis mellifera*) were tested. Add also how many bees were included into one sample.

Author response:

For this study, a total of seven *Apis mellifera* pooled samples were tested by RNA sequencing. Each sample was a pool of total RNA from five individual bees. Additionally, one publicly available *A. mellifera* dataset (SRR28408579) was included, bringing the total to eight *A. mellifera* datasets analyzed for viral detection and diversity. These details are described in full from lines 406 to 430.

5. Page 19, line 445-check one space in bracket

Author response:

Checked and corrected

6. Page 20, line 461-469, check some words are in bold

Author response:

Checked and corrected

Reviewer #2 (Comments for the Author):

This manuscript provides valuable descriptive insights into the virome of both managed and native bees in Bangladesh using RNA sequencing. The topic is timely and relevant, and the work contributes to the growing body of regional pollinator virome studies. The manuscript is generally well written and clear. I have a few comments for the authors to consider:

1. Bias in methodology

The study relies on poly(A)-selected RNA sequencing. This inherently enriches for polyadenylated RNA viruses and excludes some viral groups. Consequently, statements such as "Using high-throughput RNA sequencing, we show that viruses of the order Picornavirales dominate the viral landscape of both western and native bees" (lines 24-26) should be moderated, as this conclusion may reflect methodological bias as much as true biological dominance. Please consider rephrasing these claims to more accurately reflect the scope of the data.

Author reply:

We have revised the sentences to more accurately reflect the scope and limitations of our data. The revised text now reads: "Using high-throughput poly(A)-selected RNA sequencing, we observed that viruses of the order *Picornavirales* are frequently detected in both western and native bees. However, this pattern may reflect both true biological abundance and methodological bias, as this approach inherently enriches for polyadenylated RNA viruses."

2. Novel virus genomes and data availability

While raw RNA-seq reads have been deposited, several new or unreported viral genomes are described in the manuscript (e.g., novel Iflaviridae and Dicistroviridae). If the viruses are near-complete it might be beneficial to describe them further and deposit them in GenBank and provide the accession in Data availability section.

Author reply:

We thank the reviewer for this valuable suggestion. The novel viral genomes described in our manuscript represent partially or near-complete assemblies, and additional validation is needed to confirm genome termini and completeness. As requested, we have provided all known and novel virus sequences identified as supplementary material and deposited the raw RNA-seq reads in a public repository. Please check line 535 to 539. The primary objective of this study was to explore the occurrence and diversity of both known and novel viruses in bee species from Bangladesh, which does not permit detailed molecular or biological descriptions of the newly identified viruses. However, in line with the reviewer's constructive suggestion, we plan to design a future project to address this in more depth.

3. Choice of assembler

Trinity is a standard tool for transcriptome assembly and is used in virome studies, but viruses are not transcripts, and metagenomic assemblers (e.g., MEGAHIT, metaSPAdes) may perform better in reconstructing viral genomes, particularly in mixed samples. While Trinity is acceptable here in combination with stringent QC, it may be useful to consider alternative assembly tools in future studies.

This is a well-executed descriptive study that will add to the global picture of bee viromes.

The manuscript would benefit from tempering some claims, clarifying data deposition, and

more explicitly acknowledging methodological constraints (poly-A bias, sampling unevenness).

Author reply:

We thank the reviewer for this insightful comment and for recognizing the value of our descriptive study. To address potential limitations of Trinity, we implemented several strategies, including stringent quality control measures (read trimming, removal of host sequences, and contig validation), cross-checking assembled contigs against reference viral databases to minimize misassemblies, and confirming the presence of key viral sequences across multiple samples to ensure reproducibility, as detailed in the manuscript (Lines 436–459). We acknowledge that metagenomic assemblers such as MEGAHIT or metaSPAdes may offer advantages for reconstructing mixed viral populations, particularly low-abundance or segmented viruses, and we will explore these tools in future work. In response to reviewer 2's suggestions, we have revised the manuscript by tempering claims regarding viral dominance (Lines 24–26, 265, 282–285), clarifying data deposition to indicate that raw RNA-seq reads and novel virus sequences are provided as supplementary material (Lines 534–539, Data accessibility section), and explicitly acknowledging methodological constraints, including poly(A)-selection bias and uneven sampling (Lines 24–28, 166–169, 282–285, 415–417).

Re: Spectrum01971-25R1 (**Insights into the Viral Landscape of the Western Honey Bee and Native Bees in Bangladesh**)

Dear Dr. Islam Hamim:

Your manuscript has been accepted, and I am forwarding it to the ASM production staff for publication. Your paper will first be checked to make sure all elements meet the technical requirements. ASM staff will contact you if anything needs to be revised before copyediting and production can begin. Otherwise, you will be notified when your proofs are ready to be viewed.

Sincerely,
Jonathan Snow
Editor
Microbiology Spectrum